# Defining the therapeutic selective dependencies for distinct subtypes of PI3K pathway-altered prostate cancers

Ninghui Mao[1,10], Zeda Zhang[1,2,10], Young Sun Lee[1], Danielle Choi[1], Aura Agudelo Rivera[1], Dan Li [1], Cindy Lee[1], Samuel Haywood[3,4], Xiaoping Chen[5], Qing Chang[5], Guotai Xu[1], Hsuan-An Chen [2], Elisa de Stanchina[5], Charles Sawyers [1,6], Neal Rosen [7], Andrew C. Hsieh [8], Yu Chen [1,9] & Brett S. Carver [1,3,4 ✉]

Previous studies have suggested that *PTEN* loss is associated with p110β signaling dependency, leading to the clinical development of p110β-selective inhibitors. Here we use a panel pre-clinical models to reveal that PI3K isoform dependency is not governed by loss of *PTEN* and is impacted by feedback inhibition and concurrent *PIK3CA/PIK3CB* alterations. Furthermore, while pan-PI3K inhibition in *PTEN*-deficient tumors is efficacious, upregulation of Insulin Like Growth Factor 1 Receptor (IGF1R) promotes resistance. Importantly, we show that this resistance can be overcome through targeting AKT and we find that AKT inhibitors are superior to pan-PI3K inhibition in the context of *PTEN* loss. However, in the presence of wild-type *PTEN* and *PIK3CA*-activating mutations, p110α-dependent signaling is dominant and selectively inhibiting p110α is therapeutically superior to AKT inhibition. These discoveries reveal a more nuanced understanding of PI3K isoform dependency and unveil novel strategies to selectively target PI3K signaling nodes in a context-specific manner.

[1] Human Oncology and Pathogenesis Program, Memorial Sloan Kettering Cancer Center, New York, NY, USA. [2] Louis V. Gerstner Jr. Graduate School of Biomedical Sciences, Memorial Sloan Kettering Cancer Center, New York, NY, USA. [3] Department of Surgery, Memorial Sloan Kettering Cancer Center, New York, NY, USA. [4] Department of Urology, Memorial Sloan Kettering Cancer Center, New York, NY, USA. [5] Antitumor Assessment Core Facility, Memorial Sloan Kettering Cancer Center, New York City, NY, USA. [6] Howard Hughes Medical Institute, Memorial Sloan Kettering Cancer Center, New York, NY, USA. [7] Molecular Pharmacology Program, Memorial Sloan Kettering Cancer Center, New York, NY, USA. [8] Divisions of Human Biology and Clinical Research, Fred Hutchinson Cancer Research Center, Seattle, WA, USA. [9] Department of Medicine, Memorial Sloan Kettering Cancer Center, New York, NY, USA. [10]These authors contributed equally: Ninghui Mao, Zeda Zhang. ✉email: carverb@mskcc.org

The phosphoinositide 3-kinase (PI3K) pathway is one of the most altered oncogenic pathways in human cancer[1,2]. In hormone-regulated cancers such as prostate and breast cancer, activation of the PI3K pathway is often driven by mutations/amplification/overexpression of *PIK3CA/PIK3CB*, or loss of the tumor suppressor *PTEN*[3–6]. According to recent cancer genomic sequencing efforts, *PIK3CA* alterations are found in more than 30% of breast cancers, and *PIK3CA/PIK3CB* alterations are observed in 15% of prostate cancers[3–6]. Loss of *PTEN* occurs in approximately 40% of metastatic prostate cancers and represents the most common alteration in the PI3K pathway[3,4]. Previous studies using preclinical cell lines and xenograft models have shown that targeting PI3K activity could be a promising therapeutic strategy to treat tumors with PI3K pathway alterations[7,8]. However, early attempts to develop PI3K inhibitors did not advance clinically due to nonselective target inhibition, toxicity necessitating dose reductions, and the recently described concept of relief of feedback inhibition, all of which reduced efficacy[9,10].

The PI3K holoenzyme is composed of a catalytic subunit (p110) and a regulatory subunit (p85, p55, and p50). In epithelial lineages, p110α (*PIK3CA*) and p110β (*PIK3CB*) are the dominantly expressed catalytic subunit isoforms[8,11]. Oncogenic activity of PI3K-altered tumors is often dependent on different p110 isoforms[12–16]. In breast cancer, activating *PIK3CA* mutations are dominant drivers of PI3K signaling whereas in *PTEN*-deficient prostate cancer, such activity is reported to be driven by p110β[16,17]. These findings subsequently propelled the development of the next generation of agents designed to target individual p110 isoforms in a selective manner[18–20].

Previous studies have documented a reciprocal negative feedback regulation between PI3K activity and hormone receptor (e.g. androgen receptor) signaling in hormone-dependent cancers including breast and prostate cancer[21–24]. This work provided the molecular rationale for the development of clinical trials co-targeting hormone receptor and PI3K signaling in PI3K pathway-altered breast or prostate tumors. A phase II trial of combining a second-generation AR antagonist and an inhibitor of AKT, a downstream component of the PI3K pathway, demonstrated significant improvement in radiographic progression-free survival (RPFS) for metastatic prostate cancer patients with tumors displaying loss of *PTEN*[25]. The results of this trial have led to the accrual of a phase III study that recently met its co-primary endpoint in *PTEN*-loss metastatic castration resistant prostate cancer (mCRPC) demonstrating a significant improvement in RPFS. Additionally, the recent phase III SOLAR1 trial reported a significant improvement in progression-free survival for breast cancer patients harboring *PIK3CA* mutations when treated with BYL719 (p110α inhibitor) plus fulvestrant (ER degrader)[14,26–28]. In prostate cancer, while previous studies suggest that loss of *PTEN* is associated with p110β dependency and early phase clinical trials have been developed to evaluate the efficacy of p110β-selective inhibitors in this context, several caveats to this approach remain. First, the original studies reporting that loss of *PTEN* promoted p110β dominant signaling were performed in a limited number of models. Secondly, investigators have shown that in the context of *PTEN* loss, inhibition of p110β leads to relief of feedback inhibition of the non-dominant p110 isoforms resulting in PI3K rebound signaling and diminished efficacy[29]. Despite these caveats, enthusiasm for p110 isoform-selective inhibitors for the management of PI3K pathway-altered prostate cancer continues, with the hope of maximizing efficacy and minimizing toxicity.

Here, we analyze a broad collection of prostate cancer models to identify the appropriate molecular context and therapeutic strategies to selectively target PI3K pathway-altered prostate cancers. We define the impact of *PTEN* status and PI3K alterations on p110 isoform dependency, further study the role of relief of feedback between p110 isoforms across the common PI3K pathway alterations and reveal a novel and actionable mechanism of resistance to PI3K inhibition.

## Results

**Loss of *PTEN* does not drive p110β dependency.** Alterations in the PI3K pathway (*PTEN*, *PIK3CA*, *PIK3CB*, and *PIK3R1*) are seen in more than 50% metastatic castration-resistant prostate cancer (mCRPC) (Fig. 1a). Among these cases, the majority (82%) harbor *PTEN* deletion or mutation, while others (~11–19%) display amplification, mutations, or mRNA upregulation in *PIK3CA* or *PIK3CB* (Fig. 1b). Additionally, further characterization reveals a significant co-occurrence between *PTEN* and *PIK3CA/PIK3CB/PIK3R1* alterations despite their epistatic relationship (Fig. 1c). This high rate of co-occurrence indicates that concomitant alterations in members of the PI3K pathway may have additive or synergistic roles in prostate tumorigenesis.

To systematically model PI3K pathway alterations and study p110 isoform dependencies in mCRPC, we evaluated a broad collection of patient-derived prostate cancer organoids (PDOs) established from metastatic tumor biopsies of mCRPC patients[30]. We also analyzed two previously established prostate cancer cell lines, LNCaP (*PTEN* loss) and CWR22Pc (*PIK3CA* Q546R). These prostate cancer models harbor various alterations in the PI3K pathway and exhibit differential PI3K signaling activity, as shown by the levels of downstream AKT phosphorylation (Fig. 1d, e). Focusing on the common regulators of PI3K signaling, we observed a high frequency of genomic deletion of *PTEN* across the pre-clinical models, one cell line with a *PIK3CA* activating mutation, and three with alterations of PIK3R1. While *PIK3CB* and *AKT* activating mutations occur at a low frequency in mCRPC, we did not identify alterations or activating mutations of these genes in our models.

PI3K signaling in *PTEN*-deficient prostate cancer has been reported to be dependent on p110β, but the models evaluated were limited in number[12,13,15]. Using our collection of PDOs, we first tested if their PI3K activity is dependent on p110α or p110β by treating them with a p110α-selective inhibitor (BYL719) or a p110β-selective inhibitor (AZD8186). We used both compounds at their on-target dosage that does not cross-inhibit other PI3K isoforms[1]. Surprisingly, despite their *PTEN* deficiency, MSK-PCa3, MSK-PCa11, and MSK-PCa8 showed selective sensitivity to p110α inhibition (BYL719) rather than p110β inhibition (AZD8186), while MSK-PCa1 displayed p110β dependency similar to LNCaP (Fig. 1f). Since organoid culture media contains various growth factors, which may bias PI3K activation through receptor tyrosine kinase (RTK)-mediated p110α activation, we next tested if p110α or p110β dependency is influenced by the components in media. We switched the regular 10% FBS RPMI media with organoid culture media for 48 h and observed that PI3K isoform dependency remained the same in both p110β-dependent LNCaP and p110α-dependent CWR22Pc (Supplementary Fig. 1A, B). Furthermore, we evaluated the concept that *PTEN* loss causes p110β dependency in prostate cancer. To accomplish this, we performed CRISPR-Cas9 knockout of *PTEN* in two *PTEN* wild-type prostate cancer organoid models and the *PTEN* wild-type CWR22Pc cell line. When *PTEN* was deleted we observed that all three models remain predominantly dependent on p110α, not on p110β (Fig. 1g and Supplementary Fig. 1C). Conversely, to evaluate the effect of *PTEN* expression on p110β dependency, we ectopically expressed wild-type *PTEN* in LNCaP cells using a doxycycline-inducible Tet-on system. Indeed, we

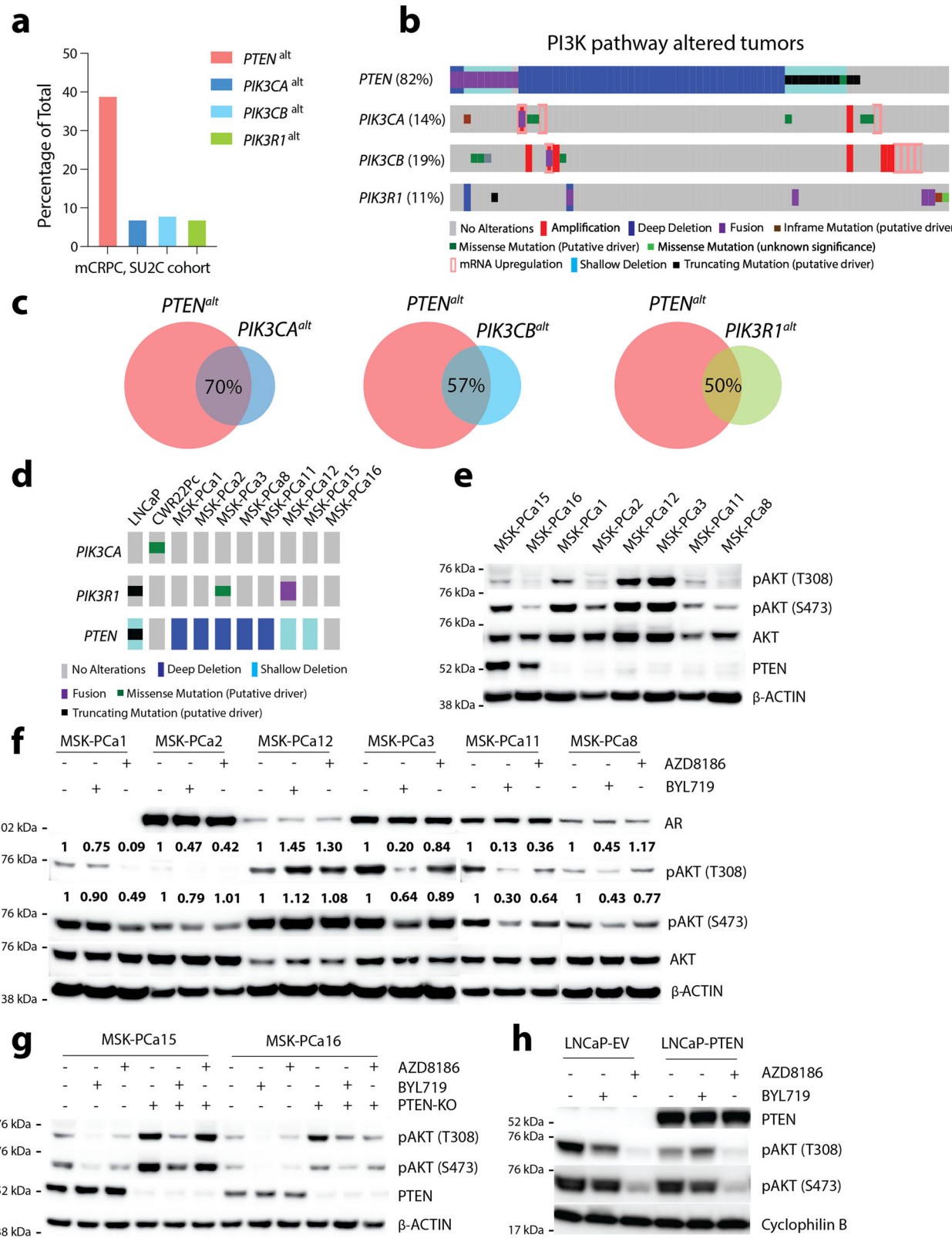

detected a reduction of PI3K activity, however, those LNCaP-PTEN cells retained their inherent p110β dependency (Fig. 1h). Collectively, using ex vivo PDOs and in vitro cell line models, we showed that p110α or p110β dependency is not solely determined by PTEN status. These results highlight that loss of PTEN is not a dominant biomarker for p110β-selective signaling and raises the need to further define the mechanisms that drive p110 isoform

dependency. This discovery carries clinical significance as the development of p110β isoform-selective inhibitors has been focused on cancers harboring loss of PTEN.

**Activating _PIK3CA_/_PIK3CB_ mutations determine p110 isoform dependency in prostate cancer.** We next sought to investigate if there is a clinically relevant mechanism that could shape

**Fig. 1 Modeling PI3K alterations using patient-derived organoid (PDO) reveals PI3K isoform dependency is not determined by PTEN. a** Percentage of PI3K members (*PTEN, PIK3CA, PIK3CB,* and *PIK3R1*) alterations in metastatic castration-resistant prostate cancer cohort (SU2C Cell 2015). **b** Oncoprint showing PI3K member alterations in all PI3K-altered tumors from the same cohort (*n* = 76). **c** Venn diagram showing the percentage of overlaps among alterations in *PIK3CA, PIK3CB, PIK3R1,* and *PTEN*. **d** Oncoprint showing *PIK3CA, PIK3R1,* and *PTEN* alterations in a collection of PDOs. **e** Western blot showing pAKT level and *PTEN* status in a collection of PDOs. **f** Western blot showing AR and pAKT levels in PDOs treated with BYL719 (1 μM), AZD8186 (250 nM), or Veh for 4 h. Quantification of pAKT473 and pAKT308 normalized to β-actin. **g** Western blot showing pAKT levels in isogenic PDO pairs MSK-PCa15 (*PTEN*-intact, *PTEN*-KO) and MSK-PCa16 (*PTEN*-intact, *PTEN*-KO) treated with BYL719 (1 μM), AZD8186 (250 nM), or Veh for 4 h. **h** Western blot showing pAKT levels in isogenic LNCaP-EV and LNCaP-*PTEN* pairs treated with doxycycline (500 ng/mL) for 8 h for *PTEN* induction, followed by either BYL719 (1 μM), AZD8186 (250 nM), or Veh for 4 h. All assays were performed with three biological replicates. Source data are provided as Source data file.

p110 isoform dependency in prostate tumors. By analyzing whole-exome sequencing results from an advanced prostate cancer patient cohort, we explored potentially activating alterations in the PI3K pathway and identified oncogenic alterations across the dominant drivers of PI3K signaling (*PTEN, PIK3CA, PIK3CB, PIK3R1*) (Fig. 1a). Interestingly, while activating alterations in *PIK3CA/PIK3CB* can occur in the setting of wild-type *PTEN*, the alterations were more frequently enriched in *PTEN*-altered tumors (Fig. 1b, c). Based on this observation, we hypothesized that gain-of-function alterations in *PIK3CA* or *PIK3CB* could work synergistically with *PTEN* deficiency to further enhance oncogenic PI3K signaling and shape p110 isoform dependency. To this end, we expressed p110α^E545K or p110β^E1051K activating mutants in the pre-clinical models (LNCaP, CWR22Pc, and MSK-PCa1) using lentiviral transduction. Ectopic expression of mutant p110α^E545K or p110β^E1051K further enhanced baseline PI3K activity in both the *PTEN*-null (LNCaP, MSK-PCa1) and *PTEN* wild-type (CWR22Pc) backgrounds (Fig. 2a–c). These results suggest that hyper-activation of p110 subunits can further elevate PI3K pathway activity regardless of *PTEN* status and there is cooperativity among activating alterations in members of the PI3K pathway. We then sought to ask if p110α/p110β-activating mutations influence p110 isoform dependency in the *PTEN*-deficient model LNCaP. Surprisingly, we observed that expression of p110α^E545K in LNCaP cells shifted their p110β dependency towards more p110α-dependent signaling (Fig. 2d). LNCaP-p110α^E545K cells were also no longer sensitive to AZD8186 (p110β inhibition) at its on-target or higher concentration dosing in in vitro growth assays (Fig. 2e, Supplementary Fig. 2A). Importantly, while standard dosing of BYL719 (p110α inhibition) was not sufficient to inhibit cell proliferation, to account for the exogenous p110α^E545K overexpression, we indeed found that increasing the concentration of BYL719 suppressed cell growth (Fig. 2e, Supplementary Fig. 3A). Similarly, in the *PTEN* wild-type model CWR22Pc, expression of p110β^E1051K shifted its p110α dependency towards more p110β-dependent signaling (Fig. 2f, g and Supplementary Fig. 2B, C). Additionally, we found that in the context of *PTEN* loss the overexpression of the activating PIK3CA/PIK3CB mutants did not affect cell proliferation, presumably because PI3K signaling is already robustly activated. However, in the setting of wild-type *PTEN*, expression of these mutants significantly promoted cell proliferation (Fig. 2e, g).

Given that some of the *PIK3CA/PIK3CB* alterations involve amplification or mRNA upregulation of wild-type alleles, we evaluated if ectopic expression of wild-type *PIK3CA* impacted PI3K signaling. Indeed, overexpression of p110α^WT in a *PTEN* deficient (p110β-dependent) pre-clinical model promoted PI3K signaling through p110α, impacting isoform selective signaling dependency. However, this effect was not as robust as the expression of constitutive activating *PIK3CA* mutations (Fig. 2h). This suggests that while overexpression of wild-type p110α can promote isoform selective signaling and dependency, activating *PIK3CA* mutations indeed play a more dominant role. Taken

together, we have demonstrated that aberrant PI3K signaling driven by amplification or activating mutations of *PIK3CA* or *PIK3CB* are able to shift p110 isoform dependency regardless of *PTEN* status across multiple prostate cancer models.

**PTEN-deficiency enhances relief of feedback inhibition through p110α and p110β signaling following contralateral isoform selective inhibition.** Previous studies in prostate cancer harboring loss of *PTEN* have demonstrated the impact of relief of feedback inhibition and activation of p110α following p110β-selective inhibition[31,32]. Furthermore, recent results suggested that in *PIK3CA* or *HER2*-mutant breast cancers, loss of *PTEN* was a driver of resistance to p110α-targeting strategies through the maintenance of PI3K signaling by alternative p110 isoforms[33]. To determine if relief of feedback inhibition was reciprocal between p110α and p110β, we performed time-course experiments, using two p110β-dependent models, MSK-PCa1 and LNCaP, and a p110α-dependent model, MSK-PCa3. Similar to our previous studies in LNCaP, following the initial potent response to PI3K signaling inhibition with AZD8186 (p110β inhibitor), reactivation of PI3K signaling occurred through p110α in MSKPCa1 and LNCaP (Fig. 3a and Supplementary Fig. 3D)[31]. Importantly, for the first time, we were able to show that in a *PTEN*-deficient prostate cancer model that displays p110α dominance (MSK-PCa3), selective inhibition of p110α (BYL719) resulted in feedback activation of PI3K signaling through p110β (Fig. 3c). Feedback activation of PI3K was effectively blocked with dual inhibition of both p110 isoforms. Combined inhibition of p110α + p110β also achieved the most potent anti-proliferative effect in three independent PDO and cell line models in vitro (Fig. 3b, d and Supplementary Fig. 3C). Our data demonstrate that in the context of *PTEN* loss, regardless of p110 isoform signaling dominance, targeting both p110α and p110β is required to overcome relief of feedback inhibition.

To determine if *PTEN* status would affect the modulation of feedback, we next used the isogenic pairs of CWR22Pc-sg*NT* and -sg*PTEN* to test the combinatorial effect of p110α and p110β dual inhibition. In these *PIK3CA*-mutant, p110α-dependent isogenic pairs, we observed a significantly more robust rebound of PI3K signaling in CWR22Pc-sg*PTEN* cells as compared to the *PTEN*-intact isogenic pair (CWR22Pc-sg*NT*) following p110α inhibition (Fig. 3e, f). In both CWR22Pc-sg*NT* and -sg*PTEN* cells, potent PI3K signaling inhibition was achieved by co-targeting p110α and p110β (Fig. 3e, f). However, despite combined selective p110α and p110β inhibition, low levels of rebound signaling through pAKT remains in CWR22Pc-sg*PTEN*, indicating an incomplete PI3K signaling inhibition even with the combination (Fig. 3f). Furthermore, in our *PTEN* deficient prostate cancer models (LNCaP, MSKPCa1, and MSKPCa3) a similar phenotype of residual rebound PI3K signaling is observed following combined p110α and p110β inhibition (Fig. 3a, c and Supplementary Fig. 3D). Collectively, this data suggests that targeting AKT directly may be the optimal approach for suppressing downstream PI3K signaling in the molecular context of loss of *PTEN*.

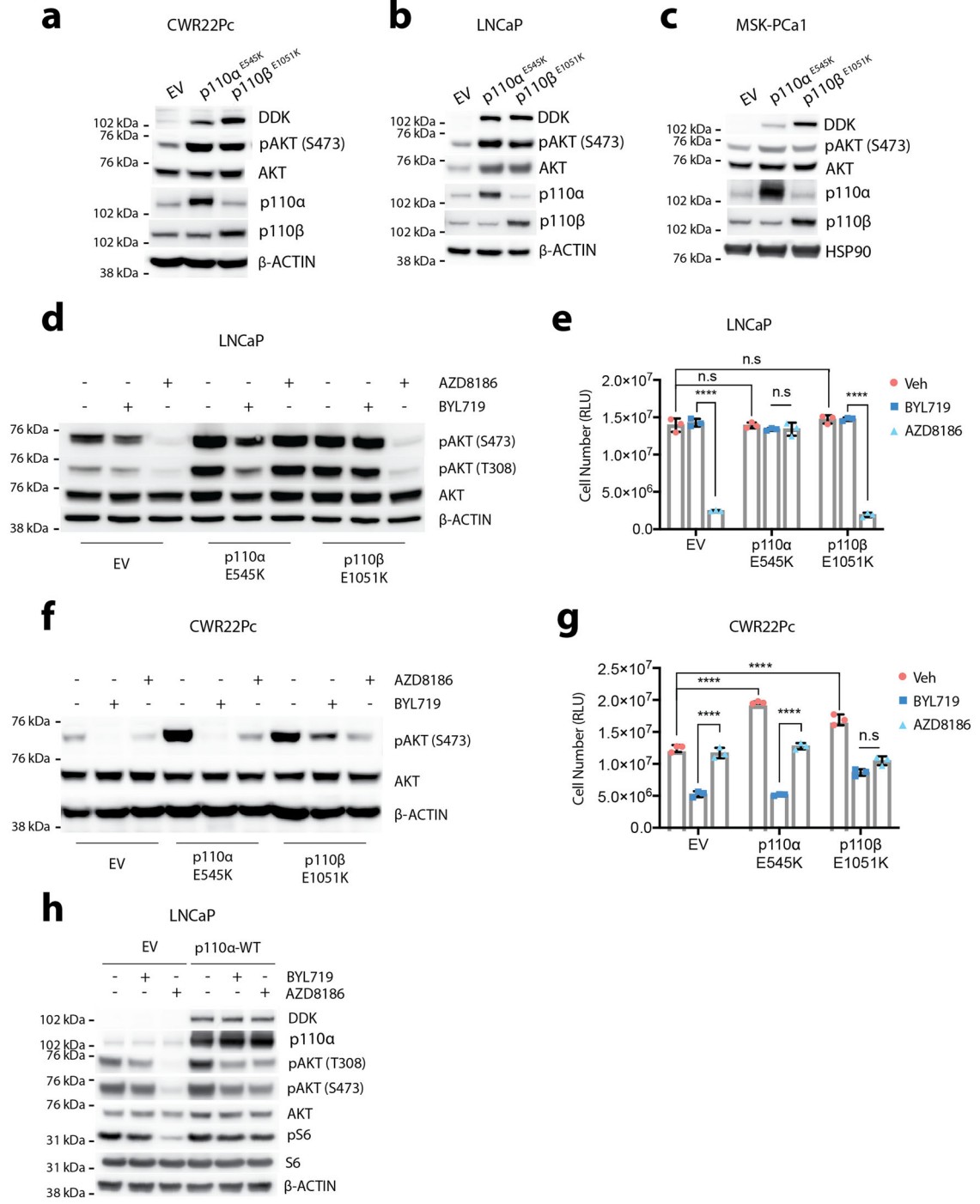

**Fig. 2 Activating *PIK3CA/PIK3CB* mutations determine p110 isoform dependency in prostate cancer. a–c** Western blot showing pAKT levels in CWR22Pc, LNCaP cells, or MSK-PCa1 organoids with ectopic expression of DDK-tagged p110α$^{E545K}$ or p110β$^{E1051K}$ mutants. **d** Western blot showing pAKT levels in LNCaP-EV, LNCaP-p110α$^{E545K}$, or LNCaP-p110β$^{E1051K}$ cells treated with BYL719 (1 μM), AZD8186 (250 nM), or Veh for 4 h. **e** CellTiter-Glo assay showing cell viability in LNCaP-EV, LNCaP-p110α$^{E545K}$, or LNCaP-p110β$^{E1051K}$ cells treated with BYL719 (1 μM), AZD8186 (250 nM), or Veh for 7 days. **f** Western blot showing pAKT levels in CWR22Pc-EV, CWR22Pc-p110α$^{E545K}$, or CWR22Pc-p110β$^{E1051K}$ cells treated with BYL719 (1 μM), AZD8186 (250 nM), or Veh for 4 h. **g** CellTiter-Glo assay showing cell viability in CWR22Pc-EV, CWR22Pc-p110α$^{E545K}$, or CWR22Pc-p110β$^{E1051K}$ cells treated with BYL719 (1 μM), AZD8186 (250 nM), or Veh for 7 days. **h** Western blot showing levels of pAKT and pS6 in LNCaP-EV (empty vector) or LNCaP-p110α$^{WT}$ (ectopic expression of DDK-tagged p110α$^{WT}$) cells cultured in regular 10% FBS-RPMI treated with BYL719 (1 μM), AZD8186 (250 nM), or Veh for 4 h. Source data are provided as Source data file. All assays were performed with three biological replicates. ****$p < 0.0001$, n.s: not significant, **e**: one-way ANOVA compared to BYL719 group, **g**: one-way ANOVA compared to AZD8186 group, error bar represents mean values ± SD.

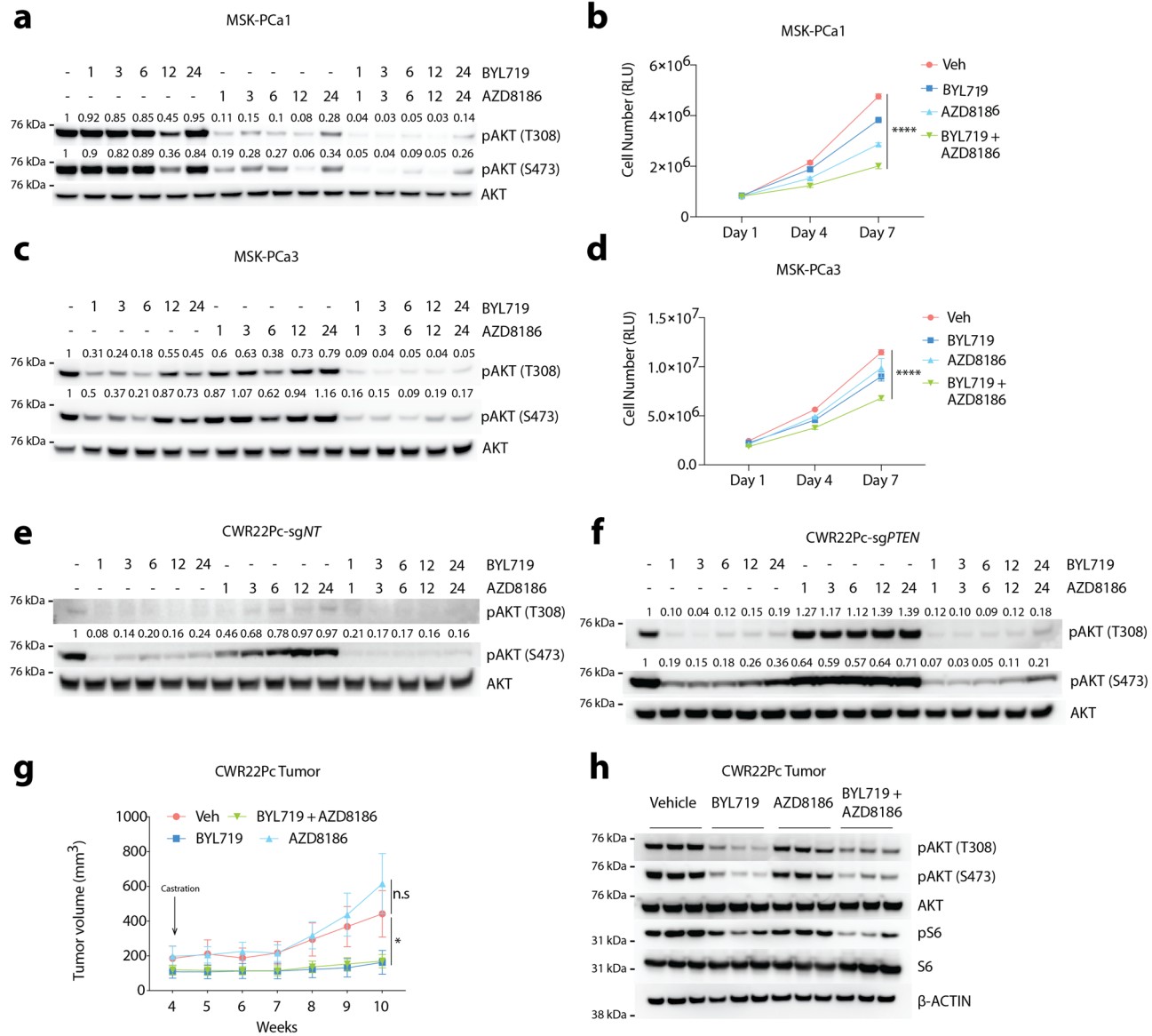

**Fig. 3 PTEN-deficiency enhances relief of feedback inhibition through p110α and p110β signaling following contralateral isoform selective inhibition.**
**a**, **c** Western blot showing pAKT levels in PDO models MSK-PCa1 and MSK-PCa3 treated with BYL719 (1 μM), AZD8186 (250 nM), BYL719 + AZD8186, or Veh for 0, 1, 3, 6, 12, and 24 h, respectively. Quantification of pAKT473 and pAKT308 normalized to total AKT. **b**, **d** Growth assay of PDO lines MSK-PCa1 and MSK-PCa3 treated with BYL719 (1 μM), AZD8186 (250 nM), BYL719 + AZD8186, or Veh. Cell number was read using CellTiter-Glo assay on Day 1, Day 4, and Day 7, respectively. **e**, **f** Western blot showing pAKT levels in CWR22Pc-sgNT or CWR22Pc-sgPTEN isogenic pairs treated with BYL719 (1 μM), AZD8186 (250 nM), BYL719 + AZD8186, or Veh for 0, 1, 3, 6, 12, and 24 h, respectively. Quantification of pAKT473 and pAKT308 normalized to total AKT. **g** Growth of CWR22Pc tumors in SCID mice treated with BYL719 (25 mg/kg), AZD8186 (75 mg/kg), BYL719 + AZD8186, or Veh. Mice were castrated when tumor reaches ~200 mm³. **h** Western blot showing pAKT and pS6 levels in CWR22Pc tumors from (**g**). Source data are provided as Source data file. All assays were performed with three biological replicates. ****$p < 0.0001$, *$p = 0.0142$, n.s: not significant, **b**, **d**, **g**: one-way ANOVA compared to Veh group, error bar represents mean values ± SD, **g**: error bar represents mean values ± SEM.

Despite the dramatic difference in PI3K rebound signaling between the two models, combined inhibition of p110α + p110β reduced proliferation in vitro compared to single isoform inhibition in both models (Supplementary Fig. 3G, H). However, in vivo studies of *PTEN* wild-type, *PIK3CA*-mutant CWR22Pc demonstrated that these models are exquisitely sensitive to castration + p110α inhibition, with no added therapeutic benefit to combined p110β inhibition, likely due to the lack of robust relief of feedback inhibition when *PTEN* is intact (Fig. 3g, h). This data demonstrate that loss of *PTEN* promotes relief of feedback inhibition and rebound signaling through alternative p110 isoforms and that in the setting of wild-type *PTEN* relief of

feedback inhibition is minimal. Taken together, these findings have important clinical implications, as approximately 5% of metastatic CRPC patients harbor activating *PIK3CA* alterations in the setting of wild-type *PTEN* and thus may benefit from p110α-specific inhibition combined with AR targeting therapies, similar to that reported in breast cancer patients in the SOLAR1 Trial.

Given the frequent co-occurrence of *PIK3CA/PIK3CB* alterations and loss of *PTEN* in prostate cancer, we evaluated how these concomitant alterations impact PI3K feedback reactivation. In LNCaP-p110α$^{E545K}$, feedback PI3K reactivation was seen following treatment with BYL719 (p110α), while dual inhibition of both isoforms significantly reduced the level of pAKT (Supplementary

Fig. 3B). Meanwhile, neither single isoform-selective inhibitor at standard dosing suppressed the growth of these mutant cells, inhibition of both isoforms was able to reduce cell proliferation (Supplementary Fig. 3A). Given that p110α$^{E545K}$ was exogenously overexpressed at high levels, we also tested the efficacy of an increased concentration of BYL719 and when combined with AZD8186 resulted in the most potent anti-proliferative effect (Supplementary Fig. 3A). Similarly, in LNCaP-p110β$^{E1051K}$, feedback PI3K reactivation was observed following AZD8186 (p110β) treatment, while dual inhibition of both isoforms significantly repressed pAKT level (Supplementary Fig. 3F). In the same model, AZD8186 as a single agent was effective in suppressing growth of mutant cells probably due to the cell line's inherent dependency on p110β, but inhibition of both isoforms most effectively suppressed cell growth (Supplementary Fig. 3E).

**Upregulation of IGF1R mediates resistance to p110α + p110β dual blockade.** Across our collection of *PTEN*-deficient PDOs, we observed that some of them exhibit exquisite sensitivity to dual inhibition of both PI3K isoforms, while others displayed innate resistance (Fig. 4a, b). When we compared their genomic profiles (MSK-IMPACT), we did not find distinct differences in the common oncogenic drivers between the two groups. We then shifted our attention to differential PI3K signaling mediators including differential RTK expression which has been implicated in altering responsiveness to PI3K inhibitors[34–36]. Exploratory RTK transcriptomic analysis across our organoid models revealed that IGF1R was upregulated in our resistant PDO MSK-PCa12 as well as in a *PTEN* wild-type PDO MSK-PCa15 (Supplementary Fig. 4A, B, Supplementary Data 3). We validated this finding at the protein level across our PDO models and found a correlation between IGF1R protein level and its activity (pIGF1R) (Supplementary Fig. 4C). Importantly, we observed significantly higher IGF1R/INSR activity in the resistant line MSK-PCa12, as compared to the sensitive line MSK-PCa11 (Fig. 4c). Higher IGF1R/INSR activity was also associated with residual levels of the AKT targets pPRAS40 and downstream pS6 when p110α + p110β dual inhibition was applied (Fig. 4c). This link suggests that residual PI3K activity may contribute to resistance following p110α + p110β dual inhibition. Based on this finding, we hypothesized that genetic ablation of IGF1R may eliminate residual level of PI3K activity and re-sensitize PDOs that are resistant to dual inhibition of p110α and p110β. We first tested this hypothesis by knocking-down IGF1R in resistant MSK-PCa12 organoids using three independent shRNA hairpins against IGF1R. In comparison to sh*Renilla* control, shRNA knockdown of IGF1R reduced AKT activity in Veh-treated condition and further suppressed AKT signaling (shown by pPRAS40 and pS6 levels) when combined with p110α + p110β dual inhibition in MSK-PCa12 (Fig. 4d). More importantly, shIGF1R restored sensitivity to p110α + p110β dual inhibition in MSK-PCa12 (Fig. 4e). These findings were validated in vivo where knockdown of IGF1R synergized with PI3K inhibition in MSK-PCa12 tumor growth and pathway inhibition (Fig. 4f, g). Meanwhile, knockdown of *INSR* had a mild effect. This result is consistent with prior knowledge that *INSR* is more relevant to metabolic control rather than cell proliferation[37] (Supplementary Fig. 5A, B).

We then tested if pharmacological inhibition of IGF1R with p110α + p110β dual inhibition would achieve similar therapeutic efficacy compared to genetic knockdown. As expected, the IGF1R/INSR dual inhibitor linsitinib plus BYL719 (p110α inhibitor) + AZD8186 (p110β inhibitor) was able to further decrease PI3K signaling and repress cell proliferation in MSK-PCa12 (Supplementary Fig. 5C, D). To validate these findings, we knocked out *PTEN* in another PDO model, MSK-PCa15, which

has high levels of endogenous IGF1R. Following *PTEN* knockout, this line displayed resistance to combined p110α + p110β inhibition, and sensitivity could be restored with linsitinib (Supplementary Fig. 5E–G). We next sought to ask if pharmacologic IGF1R inhibition could restore sensitivity of PI3K inhibition in vivo in the resistant MSK-PCa12 xenograft. Initial in vivo experiments testing the combination of BYL719 + AZD8186 + linsitinib displayed significant toxicity necessitating study termination (Supplementary Fig. 5H, I), so the experimental design was changed to evaluate the combination of linsitinib with BKM120, a pan-PI3K inhibitor. Upon 14 days post treatment, we already observed a significant anti-tumor effect in the combination condition; however, significant toxicity remained, resulting in dramatic weight loss requiring study termination for mice receiving linsitinib plus BKM120 (Fig. 4h, i). Collectively, these results suggest upregulation of IGF1R mediates residual PI3K activity and promotes resistance to p110α + p110β dual inhibition in *PTEN*-deficient prostate tumors. While these pathways are both therapeutically actionable, the toxicity observed limits the use of the combination of currently available clinical grade inhibitors.

**AKT-selective inhibition overcomes resistance caused by RTK upregulation.** AKT is one of the dominant downstream signaling components of the PI3K signaling pathway[7]. However, directly targeting AKT using ATP-competitive inhibitors was challenging due to the high similarities of ATP-binding pockets among AGC kinase family members[38]. Recently, two newly developed AKT-selective inhibitors have advanced in clinical trials of several tumor types[39–43]. We chose to study the Genentech compound ipatasertib as it is currently in a phase III clinical trial for mCRPC in combination with AR pathway inhibition based on our previous work.

To characterize the effects of ipatasertib in our model systems, we performed pharmacological dynamic studies both in vitro and in vivo using the LNCaP cell line to determine the optimum dose for target inhibition. We observed sufficient target inhibition (shown by the levels of pPRAS40 and pS6) at 500 nM in vitro and at 50 mg/kg in vivo (Supplementary Fig. 6A, B). Based on the results in the LNCaP model, we hypothesized that directly targeting AKT may allow us to overcome the RTK-mediated resistance to PI3K inhibition observed in our models. Indeed, for the BYL719 + AZD8186 resistant line MSK-PCa12, ipatasertib efficiently reduced downstream PI3K-AKT activity and achieved a potent anti-proliferative effect in vitro, combined with enzalutamide (Fig. 5a, b). Also, when we tested an independent PDO model, MSK-PCa15, which displays high levels of IGF1R and becomes resistance to p110α + p110β dual inhibition upon *PTEN* deletion, the MSK-PCa15-sg*PTEN* cells similarly showed sensitivity to ipatasertib (Supplementary Fig. 6C, D). Moreover, ipatasertib combined with enzalutamide was more effective at suppressing tumor growth than dual p110α/p110β inhibition and enzalutamide in MSK-PCa12 xenograft models (Fig. 5c and Supplementary Fig. 6E).

Given the recent positive interim analysis of the phase III ipatasertib and abiraterone clinical trial in mCRPC patients with *PTEN*-loss, there has been discussion on whether AKT inhibition should be extended to patients harboring other activating alterations of the PI3K pathway, such as *PIK3CA* mutations. To address this question, CWR22Pc xenograft models were treated with ipatasertib or BYL719 with or without castration. Surprisingly, AKT inhibition combined with castration in this molecular context had minimal effect and was inferior to BYL719 + castration (Fig. 5d). Also, AKT inhibition significantly increased MAPK signaling compared with p110α inhibition (Fig. 5e). To

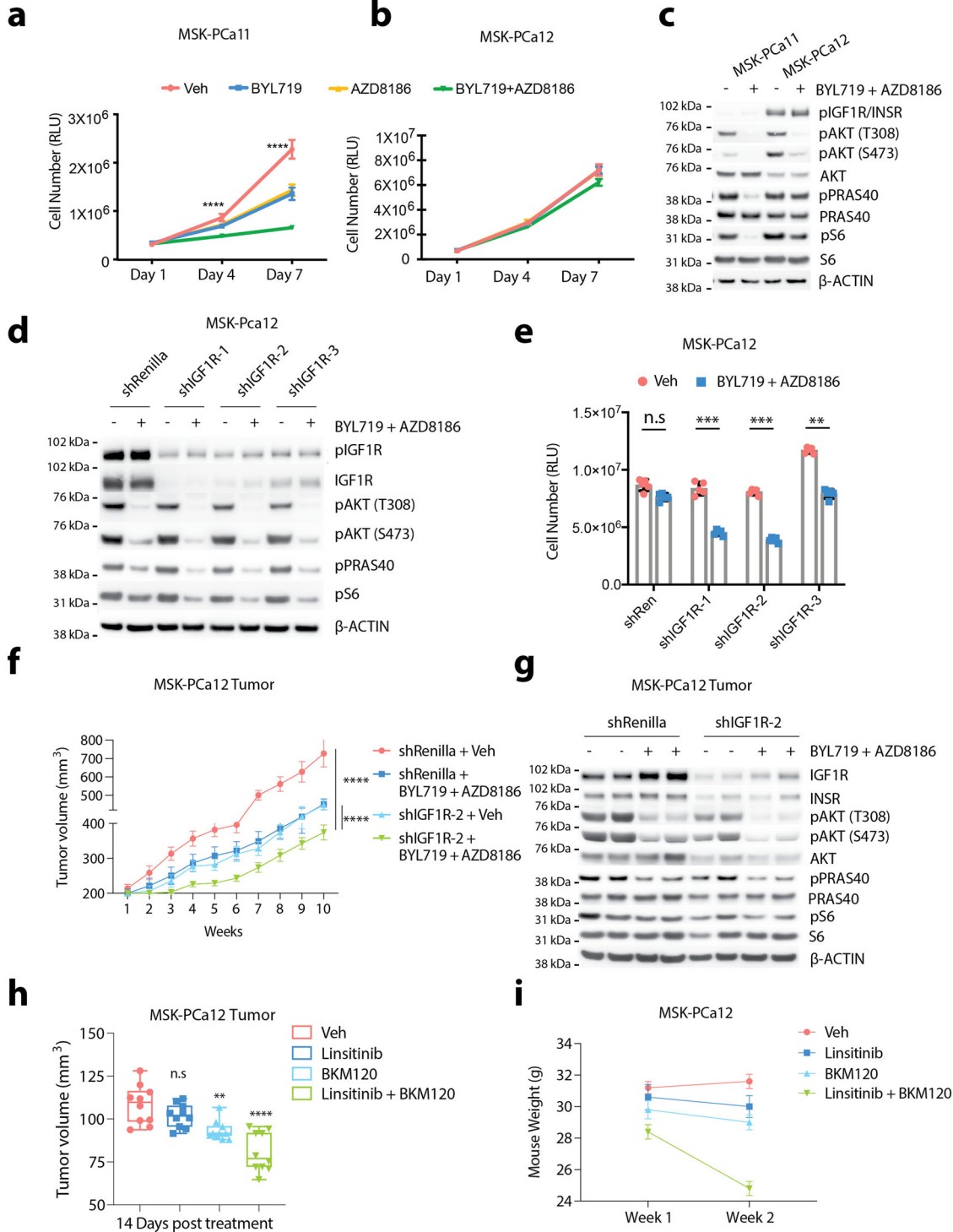

directly compare the efficacy of PI3K to AKT inhibition in hormone-regulated cancers, we evaluated a panel of prostate and breast cancer models harboring alterations in the PI3K signaling pathway. In the setting of wild-type *PTEN* where PI3K signaling was activated through *PIK3CA* mutation or RTK signaling, PI3K pathway inhibition with p110α + p110β was generally more anti-proliferative than AKT inhibition (Fig. 5f). However, in the setting of *PTEN*-loss (endogenous or isogenic *PTEN* knockout pairs), AKT inhibition showed greater efficacy at suppressing cell growth than PI3K inhibition even when p110α activating mutations were present (Fig. 5g). This association was especially significant for the breast cancer cell lines harboring mutant

*PIK3CA* when *PTEN* was exogenously knocked out (Fig. 5g, EFM19, T47D, MCF-7, MDA-MB-453, and BT20). In *PTEN*-loss, *PIK3CA* wild-type cells, AKT inhibition performed similar to PI3K inhibition with the exception of the organoids that harbored hyperactive RTK signaling where AKT inhibition was superior (Fig. 5h, MSK-PCa12, MSK-PCa15-sg*PTEN*, and MSK-PCa16-sg*PTEN*).

Collectively, our data demonstrate that in tumors harboring loss of *PTEN*, inhibiting AKT may be more efficacious than targeting PI3K, as AKT inhibitors are effective independent of RTK levels or PIK3CA mutations. Conversely, in *PIK3CA*-mutant *PTEN* wild-type prostate and breast cancers, p110α inhibition

**Fig. 4 Upregulation of IGF1R mediates resistance to p110α + p110β dual blockade. a, b** Growth assay of PDO lines MSK-PCa11 and MSK-PCa12 treated with BYL719 (1 μM), AZD8186 (250 nM), BYL719 + AZD8186 or Veh. Cell number was read using CellTiter-Glo assay on Day 1, Day 4, and Day 7, respectively. **c** Western blot showing levels of IGF1R/INSR and AKT signaling in PDO lines MSK-PCa11 and MSK-PCa12 treated with BYL719 (1 μM) + AZD8186 (250 nM) or Veh for 4 h. **d** Western blot showing levels of IGF1R and AKT signaling in MSK-PCa12-shRenilla or MSK-PCa12-shIGF1R lines treated with BYL719 (1 μM) + AZD8186 (250 nM) or Veh for 4 h. **e** CellTiter-Glo assay showing cell viability of MSK-PCa12-shRenilla or MSK-PCa12-shIGF1R lines treated with BYL719 (1 μM) + AZD8186 (250 nM) or Veh, Day 7. **f** Growth of MSK-PCa12-shRenilla or MSK-PCa12-shIGF1R tumors in SCID mice treated with BYL719 (25 mg/Kg) + AZD8186 (75 mg/Kg) or Veh ($n = 5$ mice per group). **g** Western blow showing levels of INSR, IGF1R, pAKT, pPRAS40, and pS6 in MSK-PCa12-shRenilla or MSK-PCa12-shIGF1R tumors from (**f**), 14 days post treatment. **h** Volume of MSK-PCa12 tumors in SCID mice treated with linsitinib (40 mg/kg), BKM120 (50 mg/kg), linsitinib + BKM120, or Veh, 14 days post treatment, box and whiskers Min to Max.all points ($n = 10$ mice per group). **i** Body weight analysis of mice in (**h**) ($n = 10$ mice per group). Source data are provided as Source data file. All assays were performed with three biological replicates. $****p < 0.0001$, $***p = 0.0003$, $**p = 0.0041$, n.s: not significant, **a, b**: multiple $t$-test two-sided, **e, h**: one-way ANOVA compared to Veh group, **f**: one-way ANOVA compared to shIGF1R group treated with BYL719 + AZD8186, **a, b, e**: error bar represents mean values ± SD, **f, h**: error bar represents mean values ± SEM, **i**: error bar represents ± SD.

provides superior antiproliferative effects. Furthermore, as we have previously published, due to reciprocal feedback inhibition between the PI3K and AR pathways, combined inhibition of these pathways is required in prostate cancer (Fig. 5i).

## Discussion

Targeting PI3K alterations to treat human malignancies has been proposed for more than a decade, yet previous clinical trials using PI3K pathway inhibitors against multiple human malignancies did not generate profound success due to significant toxicities and low efficacy[9,10]. Recently, improved understanding of the differential expression and functional divergence amongst PI3K isoforms raised the hope that targeting individual p110 isoforms may maximize efficacy and minimize toxicity[11,15]. For example, a p110δ-selective inhibitor idelalisib was recently approved to treat several types of blood malignancies including chronic lymphocytic leukemia, relapsed follicular lymphoma, and relapsed small lymphocytic lymphoma[44–46]. In a more recent phase III clinical trial (SOLAR-1), BYL719, a p110α-selective inhibitor combined with fulvestrant significantly improved progression-free survival in ER-positive, PIK3CA-mutant breast cancers[26]. Because PIK3CA (p110α) and PIK3CB (p110β) are ubiquitously expressed in tumors of epithelial origin[8,11], understanding p110α or p110β dependency across different genetic backgrounds is critical for selectively targeting the PI3K signaling axis in PI3K pathway-altered tumors.

In metastatic prostate cancer, PI3K pathway alterations are seen in more than 50% of patients, among which more than 80% harbor PTEN loss-of-function alterations. Limited evidence suggesting that PI3K signaling in PTEN-deficient tumors is more dependent on p110β led to the development of several phase I clinical trials evaluating p110β-selective inhibitors in PTEN-deficient mCRPC patients[14,28]. Here we characterized an expanded cohort of patient-derived organoid (PDO) models that harbor loss of PTEN to define PI3K isoform dependency. Surprisingly, we found that PTEN-status is not the determinant driver of p110 isoform selectivity. While the mechanisms determining p110 isoform dominance in PTEN-deficient prostate cancer remain unknown, this finding has important clinical implications that loss of PTEN alone will not be a biomarker of p110β inhibition sensitivity.

This unexpected finding raises the question about what molecular mechanisms drive p110 isoform selectivity. Indeed, through our genomic analyses of mCRPC, we found frequent occurrence of PIK3CA/PIK3CB/PIK3R1 alterations in the setting of both PTEN wild-type and PTEN loss. Through the modeling of PIK3CA or PIK3CB alterations using multiple PDO and cell line models, we discovered that activating mutations of p110α or p110β can drive or shift isoform dependency. This finding indicates that p110 isoform-activating mutations could be a potential

biomarker for the selection of an isoform-selective inhibitor, especially in the context of PIK3CA mutant, PTEN wild-type prostate cancer. Our results provide support for the development of clinical trials evaluating combined p110α (BYL719) and AR inhibition in patients with mCRPC harboring PIK3CA-mutant, PTEN wild-type cancers.

For the first time, we show that PI3K relief of feedback inhibition is robust and therapeutically meaningful in the context of PTEN-loss but not in the setting of wild-type PTEN. In prostate cancers harboring loss of PTEN, this feedback inhibition is reciprocal between the p110 isoforms and occurs in the context of p110α or p110β-dominant signaling. Although dual inhibition of p110α+p110β can efficiently block PI3K relief of feedback inhibition and achieve potent anti-tumor effect in some prostate cancers harboring loss of PTEN, we also identified that resistance to p110α+p110β inhibition can occur through IGF1R-mediated PI3K residual activity. While addition of an IGF1R inhibitor, linsitinib, restored p110α + p110β sensitivity in resistant lines, this combination is associated with significant toxicity limiting clinical development. Importantly, we show that the use of an AKT-selective inhibitor such as ipatasertib (GDC-0068) is not only able to overcome this resistance, but generally performs superior to pan-PI3K inhibition with lower toxicity across PTEN-deficient pre-clinical models. Moreover, these results are also supported by a recent phase II and phase III clinical trial for mCRPC patients in which treatment with ipatasertib plus abiraterone has achieved a significantly longer RPFS in PTEN-deficient patients[25]. Furthermore, in the metastatic breast cancer, PTEN alterations are shown to confer resistance to p110α-selective inhibition, demonstrating the need for an alternative strategy to target PI3K signaling[33]. Our results suggest AKT inhibitors may also be superior to pan-PI3K inhibition in breast and prostate cancers harboring loss of PTEN, however in the context of PIK3CA mutant PTEN wild-type cancers there is limited efficacy to AKT inhibition and targeting p110α is superior, secondary to dual suppression of downstream PI3K and MAPK signaling. Importantly, PIK3CA mutant PTEN wild-type prostate cancers should not be broadly grouped with PTEN loss cancers in future AKT inhibitor clinical trials. Taken together, as we gain further insights into the mechanisms driving p110 isoform dependency and validate isoform selective biomarkers, therapeutic trials may be developed to selectively target a specific isoform through a precision medicine approach.

## Methods

**Patient-derived prostate cancer organoids and cell lines.** Patient-derived organoids (PDOs) (MSK-PCa1, MSK-PCa2, MSK-PCa3, MSK-PCa8, MSK-PCa11, MSK-PCa12, MSK-PCa15, MSK-PCa16) were obtained from Dr. Yu Chen's laboratory through collaboration. MSK-PCa1, MSK-PCa2, MSK-PCa3, and MSK-PCa8 were previously published by Dr. Yu Chen's group[30]. MSK-PCa11, MSK-PCa12, MSK-PCa15, and MSK-PCa16 were newly generated and can be obtained

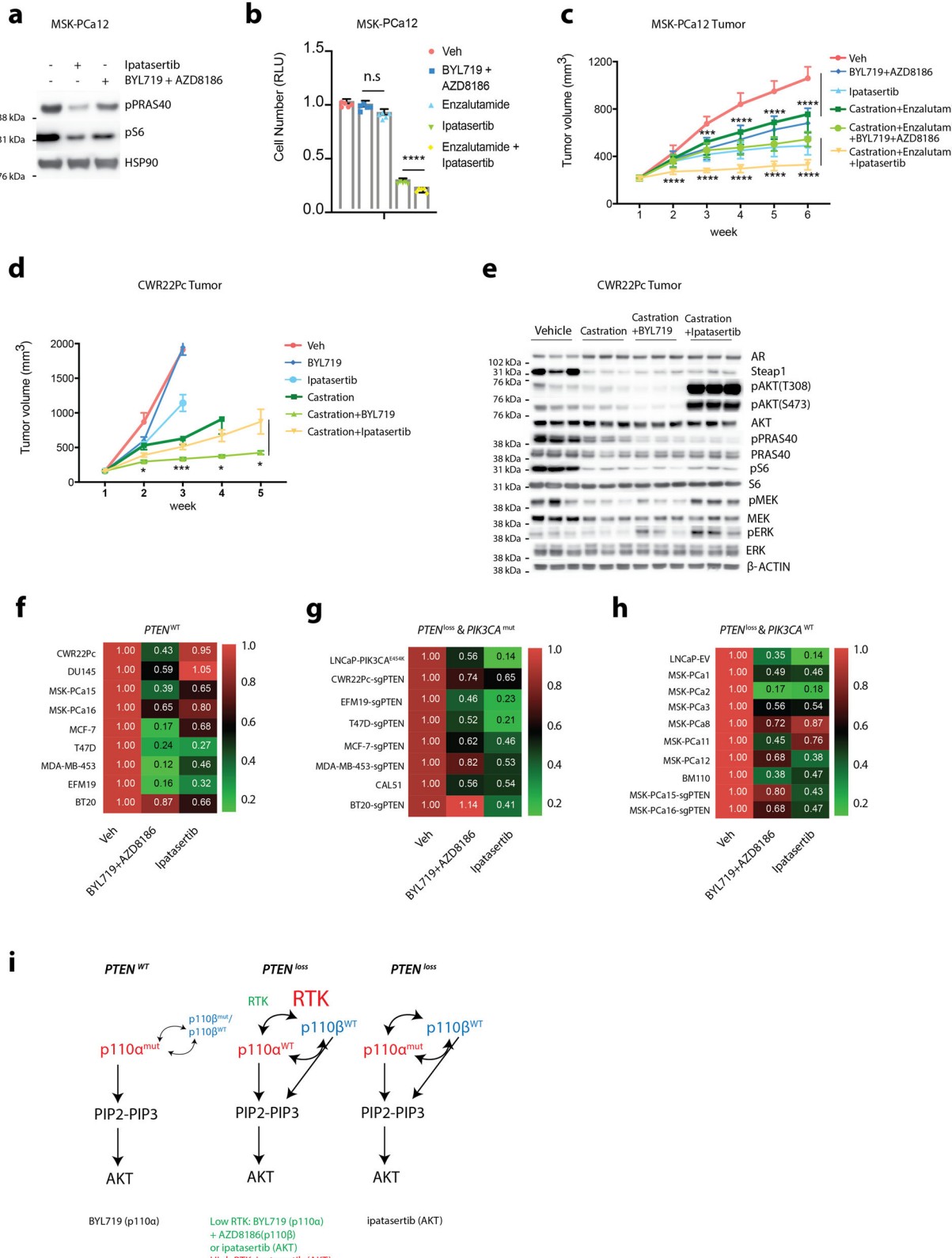

from Dr. Yu Chen's group upon request. PDOs were maintained and cultured in standard organoid culture condition[47]. LNCaP (#ATCC® CRL-1740™) and DU145 (#ATCC® HTB-81™) were obtained from American Type Culture Collection (ATCC, Manassas, VA). MCF7(#ATCC® HTB-22™), T47D (#ATCC® CRL-2865™), MDA-MB-453(#ATCC® HTB-131™), EFM19(DSMZ, #ACC-231), CAL51(DSMZ, #ACC-231) and BT20 (#ATCC® HTB-19™) cell lines were gifts from Dr. Guotai Xu (MSKCC). CWR22Pc was a gift from Marja T. Nevalainen (Thomas Jefferson University, Philadelphia, PA), and CWR22Pc-EP was generated and maintained as

previously described[48,49]. Cell lines were maintained in RPMI (LNCaP, CWR22Pc, T47D, EFM19, CAL-51) or DMEM (MCF7, MDA-MB-453, and BT20). Additional supplements for all cell lines were 10% FBS (Omega Scientific, Tarzana, CA), PenStrep (1%), L-glutamine (1%), sodium pyruvate (1%), and HEPES pH=7.6 (1%). All cell lines and prostate cancer organoids used in our studies have tested negative for mycoplasma using the MycoProbe Mycoplasma Detection Kit (R&D Systems) within one month of initiating experiments. All cell lines and organoids were freshly thawed and only passaged to achieve the number of cells required for

**Fig. 5 AKT-selective inhibition overcomes resistance caused by RTK upregulation. a** Western blot showing levels of AKT signaling in MSK-PCa12 organoids treated with ipatasertib (500 nM) or BYL719 (1 μM) + AZD8186 (250 nM) for 4 h. **b** CellTiter-Glo assay showing cell viability of MSK-PCa12 organoids treated with BYL719 (1 μM) + AZD8186 (250 nM), enzalutamide (10 μM), ipatasertib (500 nM), enzalutamide + ipatasertib, or Veh, Day 7. **c** Growth of MSK-PCa12 tumors in SCID mice treated with enzalutamide (10 mg/kg), ipatasertib (50 mg/kg), BYL719 (25 mg/kg) + AZD8186 (75 mg/kg), ipatasertib + enzalutamide, BYL719 + AZD8186 + enzalutamide or Veh. Some of mice were castrated when tumor reached ~200 mm$^3$ ($n = 5$ mice per group) ***$p$-value = 0.0011. **d** Growth of CWR22Pc tumors in SCID mice treated with BYL719 (25 mg/kg), ipatasertib (50 mg/kg) or Veh. Some of mice were castrated when tumor reaches ~200 mm$^3$ ($n = 5$ mice per group) Castration + BYL719 vs. Castration + Ipatsertinib, week 2: $p$-value = 0.045, week 3: $p$-value = 0.0086, week 4: $p$-value = 0.012, week 5: $p$-value = 0.046. **e** Western blot showing PI3K/AKT/MAPK signaling in CWR22Pc tumors from (**d**). **f–h** Heat map showing drug sensitivity of ipatasertib (500 nM) or BYL719 (1 μM) + AZD8186 (250 nM) on day 7, in cells characterized as *PTEN* wild type with *PIK3CA* mutants or RTK hyperactive (**f**), *PIK3CA* mutants with *PTEN*-loss (**g**), and *PIK3CA* wild type with *PTEN*-loss (**h**). Genetically manipulated lines are isogenic pairs. **i** Mechanism model of PI3K pathway alterations and response to clinically available therapeutic agents. All assays were performed with three biological replicates. ****$p < 0.0001$, ***$p < 0.001$, **$p < 0.01$, *$p < 0.05$, n.s: not significant, **b**: one-way ANOVA compared to Veh group, error bar represents mean values ± SD. **c**, **d**: multiple $t$-test, two-sided error bar represents mean values ± SEM.

in vitro or in vivo experiments. Cell lines and organoids developed by our group were validated by STR genotyping protocols.

**Cell proliferation assay**. $5 \times 10^3$ cells in 100 μL media were seeded per well in a 96-well plate. Drugs were administered the next day to a total volume of 200 μL of media per well, and media and drugs were replaced every 48 h to maintain their efficacy. The number of viable cells was measured using CellTiter-Glo Luminescent Cell Viability Assay kit (Promega, G7573). Luminescence was detected using a Veritas Microplate Luminometer (Turner Biosystems). Luminescence signal representing relative number of cells was recorded as RLU (relative light units) according to manufacturer's instruction. CellTiter-Glo 2.0 reagent was aliquoted into working solutions and stored at −80 °C, then at each assay time point, thawed to room temperature. Equal volume of reagent was added to wells of 96-well plates using a multi-channel pipette. Plates were incubated in room temperature on an orbital shaker for 10 m to stabilize the reaction. All cell proliferation assays were repeated in at least two independent biological replicates.

**Stable gene expression analysis**. pLenti-Lentiviral vector (pLenti-C-Myc-DDK-P2A-Neo) (Origene, PS100081) was used to express *PIK3CA* wild-type (Origene, RC213112), *PIK3CA* mutant (E545K) (Origene, RC400348), and *PIK3CB* wild-type (Origene, RC215433). *PIK3CB* mutant (E1051K) was obtained through site-directed mutagenesis with plenti-*PIK3CB* vector. The *PI3KCB* cDNA mutant was generated following manufacture's instruction using QuikChange II XL Site-Directed Mutagenesis Kit, 10 Rxn (Agilent, 200521). Inducible *PTEN* expression lentivector was obtained as a gift from Dr. Wouter Karthaus (MSKCC). Doxycycline (500 ng/mL) was added to LNCaP-EV (empty vector) or LNCaP-*PTEN* cells for 8 h for *PTEN* induction. LNCaP-EV cells serve as control. All expression vectors were sequenced verified.

**Therapeutic agents**. The AR inhibitor MDV3100, enzalutamide, was synthesized by the MSKCC chemistry core. MDV3100 was used in our studies in vitro at a concentration of 10 mmol/L and in vivo with a dose of 10 mg/kg/day[29]. The PI3K p110α-selective inhibitor BYL719 was purchased from LC Laboratories (cat# A-4477). BYL719 was dissolved in 0.5% methyl cellulose and dosed at 25 mg/kg daily P.O. The PI3K p110β-selective inhibitor AZD8186 was kindly provided by Laboratory of Neal Rosen and purchased from SelleckChem (cat# S7694). AZD8186 was dissolved in 0.5% HPMC + 0.1% Tween80 and dosed at 75 mg/kg b.i.d P.O. Pan-PI3K inhibitor BKM120 was purchased from SelleckChem (cat# S2247). BKM120 was dissolved in 0.5% methyl cellulose and dosed at 50 mg/kg daily P.O. IGF1R/INSR inhibitor linsitinib was purchased from LC Laboratory (cat# L-5814). Linsitinib was dissolved in 25 mM Tartaric Acid and dosed at 40 mg/kg daily P.O. AKT inhibitor ipatasertib was provided by Genentech under an MTA and was dissolved in 0.5% Methylcellulose and 0.2% Tween-80 for in vivo study.

**Mouse xenograft procedure**. $2 \times 10^6$ cells resuspended in 100 μL of 1:1 mix of growth media and matrigel (Corning, 356237) were engrafted or injected into 6 to 8-week-old CB17-SCID male mice (Taconic). Tumor volume was measured weekly with Peira TM900 (Peira Scientific Instruments) and was calculated using the formula: Volume 1/4 Length × Width × Height. A total of 5 mice (10 tumors) per group were used to assay tumor growth in vivo in response to treatment. Treatment was initiated when tumor volume reached ~200 mm$^3$. All experiments were approved by our IACUC protocol 06-07-012.

**Immunoblot**. Cell lysates were prepared in RIPA buffer supplemented with proteinase and phosphatase inhibitors. Proteins were resolved on NuPAGE Novex 4–12% Bis–Tris Protein Gels (Thermo Fisher Scientific) and transferred electronically onto a PVDF 0.45 mm membrane (Millipore). Membranes were blocked in 5% BSA diluted in Tris buffer saline plus 0.1% Tween 20 (TBST) for 1 h at room temperature and were incubated with primary antibodies in 5% BSA at 4 °C

overnight. After 3 washes of 10 m in TBST, membranes were incubated with secondary antibodies in 5% BSA at room temperature. All immunoblots were performed with three independent biological repeats. Representative and consistent findings were shown.

**Antibodies**. The following antibodies were used for Western blotting: AR (Abcam, ab108341), β-actin (Cell Signaling Technology, 4970S, 1:10,000), phospho-AKT (Ser473) (Cell Signaling Technology, 4060L), phospho-AKT (Thr308) (Cell Signaling Technology, 4056S), AKT (Cell Signaling Technology, 4685S), phospho-S6 ribosomal protein (Cell Signaling Technology, 4856S), S6 (5G10) ribosomal protein (Cell Signaling Technology, 2217S), *PTEN* (Cell Signaling Technology, 9188L), phospho-PRAS40 (Thr246) (Cell Signaling Technology, 2997S), PRAS40 (Cell Signaling Technology, 2691S), phospho-IGF1R (Tyr1135/1136)/INSR (Tyr1150/1151) (Cell Signaling Technology, 3024S), phospho-IGF1R (Tyr1135) (Cell Signaling Technology, 3918S), IGF1R (Cell Signaling Technology, 9750S), INSR (Cell Signaling Technology, 3025S), phospho-p44/42 MAPK (Erk1/2) (Thr202/Tyr204) (D13.14.4E) XP (Cell Signaling Technology, 4370S), p44/42 MAPK (Erk1/2) (137F5) (Cell Signaling Technology, 4695S), phospho-MEK1/2 (Ser217/221) (41G9) (Cell Signaling Technology, 9154S), MEK1/2 (D1A5) (Cell Signaling Technology, 8727S), PI3K kinase P110α (Cell Signaling Technology, 4249S), PI3K kinase P110β (Cell Signaling Technology, 3011S), HSP90 (Cell Signaling Technology, 4875S, 1:10,000), Cyclophilin B (Cell Signaling Technology, 43603S, 1:10,000), DDK tag (Origene, TA50011-100). All western blot antibodies were used at a 1:1000 dilution unless specificied.

**CRISPR/Cas9-mediated knock-out**. To knock out *PTEN* in PDOs, human prostate and breast cancer cell lines, sgRNA against human *PTEN* exon 1 was designed (https://zlab.bio/guide-design-resources/) and cloned into the Lenti-CRISPRv2 backbone (Addgene, 52962). Lentiviruses for sgRNAs were generated in 293 T cells by standard methods using Lipofectamine 2000 (Invitrogen, 11668-500). Cell lines were infected with lentivirus for 48 to 72 h and selected with puromycin (1 μg/mL) for 7 days. The target guides sequences are listed in Supplementary data 2.

**MirE-based gene knockdown**. The lentiviral (SGEP, LEPC) miR-E-based shRNA expression vectors were generously gifted by Dr. Johannes Zuber (Research Institute of Molecular Pathology, Vienna, Austria)[50]. The target guides sequences are listed in Supplementary data 2.

**RNA-Sequencing and analysis**. RNA was isolated from organoids with low passage number (less than 5) using Qiagen RNeasy kits. Library preparation and RNA sequencing were performed by MSKCC Genomics Core Laboratory using Illumina HiSeq with 50 or 75 bp paired-end reads and ~30 million reads were generated for each sample. The reads were mapped human genome reference sequence (GRC37/hg19) using STAR (v2.3)[51] and quantified into Transcripts Per Kilobase Million (TPM) using Cufflinks with upper quartile normalization (v2.1)[52]. Heatmaps analysis were generated using R-studio (package pheatmap 1.0.12). RTK normalized expression values are available in Supplementary data 1.

**Study approval**. All experiments involving animals were approved by MSKCC Animal Ethics Committee (Judith Farber).

**Reporting summary**. Further information on research design is available in the Nature Research Reporting Summary linked to this article.

## Data availability

The RNAseq data performed on the patient derived prostate cancer organoids was deposited to GEO (accession number GSE181374), and can be accessed at: https://

www.ncbi.nlm.nih.gov/geo/query/acc.cgi?acc=GSE181374. The processed RNAseq data are also available for analysis through cBioPortal For Cancer Genomics (https://www.cbioportal.org) using the link: https://www.cbioportal.org/study/summary?id=prad_mskcc_cheny1_organoids_2014.

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

## Acknowledgements

A special thanks to members of the Chen, Sawyers, and Rosen labs for providing informative discussion. This work was funded in part through NIH/NCI Prostate SPORE P50-CA092629 (B.S. Carver, N. Rosen, C.L. Sawyers, Y. Chen), NIH/NCI U54-CA224079 (B.S. Carver, Y. Chen, C.L. Sawyers), NIH/NCI R01-CA182503 (B.S. Carver, N. Mao, and Y.S. Lee), PCF Challenge Award (B.S. Carver, Y. Chen, and A.C. Hsieh), NIH/NCI R37 CA230617 (A.C. Hsieh), and the MSKCC NIH/NCI Cancer Center Support Grant P30

CA008748. Funding through the STARR Cancer Consortium (Y. Chen, B.S. Carver) allowed for establishment of a prostate organoid core to assist in our experiments. All authors approved the final manuscript. Z.Z. is supported by the NCI Predoctoral to Postdoctoral Fellow F99/K00 Transition Award (F99CA223063).

## Author contributions

N.M., Z.Z., Y.C., A.C.S., and B.S.C. conceived the project. N.M., Z.Z., N.R., C.L.S., and B.S.C. oversaw the project, performed experimental design and data interpretation. Z.Z. and B.S.C. co-wrote the manuscript. N.M., Y.L., and A.C.S. edited the manuscript. Z.Z., N.M., Y.L., D.C., A.A.R., X.C., Q.C., and E.D.S. performed in vivo experiments. Y.C., H.D., C.L., S.H., D.L., and G.X. assisted with the prostate cancer organoid model experiments. N.M., Z.Z., Y.L., and H.A.C. cloned plasmid reagents. N.M., Z.Z., and Y.L. performed western blots, in vitro growth assay, and qRT-PCR analysis. Z.Z., N.M., and B.S.C. performed analysis on cbioportal.org. All authors approved of the final manuscript.

## Competing interests

C.L.S. is co-inventors of enzalutamide and apalutamide and may be entitled to royalties. C.L.S. serves on the Board of Directors of Novartis and is a co-founder of ORIC Pharm. He is a science advisor to Agios, Beigene, Blueprint, Column Group, Foghorn, Housey Pharma, Nextech, KSQ, Petra, and PMV. He was a co-founder of Seragon, purchased by Genentech/Roche in 2014. N.R. is on the scientific advisory board (SAB) and receives research funding from Chugai, on the SAB and owns equity in Beigene, and Fortress. N.R. is also on the SAB of Daiichi-Sankyo, Astra-Zeneca-MedImmune, and F-Prime, and is a past SAB member of Millenium-Takeda, Kadmon, Kura, and Araxes. N.R. is a consultant to Novartis, Boehringer Ingelheim, Tarveda, and Foresight and consulted in the last three years with Eli Lilly, Merrimack, Kura Oncology, Araxes, and Kadman. N.R. owns equity in ZaiLab, Kura Oncology, Araxes, and Kadman. N.R. collaborates with Plexxikon. A.C.H. has received research funding from eFFECTOR Therapeutics. The remaining authors declare no competing interests.
