## [Peer Review File · Nature Communications]

Reviewers' Comments:

Reviewer #1:

Remarks to the Author:

The manuscript by Mao et al. convincingly shows that PI3K p110 isoform dependency in prostate cancer is not simply driven by loss of PTEN, and gain-of-function mutations in PIK3CA or PIK3CB can confer or drive isoform dependency. Combined inhibition of both alpha and beta p110 isoforms is superior to selective inhibition, but resistance to p110 inhibitor combination treatment can be mediated by upregulation of IGF1R.

Taken together, the authors' findings show that direct inhibition of AKT may be superior to PI3K inhibition in PTEN-deficient tumors, whilst in PTEN wild-type tumors inhibition of p110 alpha is superior, due to simultaneous suppression of MAP kinase signaling.

These findings advance our understanding of the biology of PI3K-AKT signaling in distinct subsets of prostate cancers and provide a basis to explore differential therapeutic strategies based on biomarkers that are readily quantifiable.

The experimental approach is generally robust and the statistical methods used are appropriate. Addressing the following comments will further strengthen the manuscript:

1. Figure 2H: ectopic overexpression of wild type p110alpha in LNCaP cells should be documented by Western blotting
2. Figure 3F: The text states that "In both CWR22Pc-sgNT and -sgPTEN cells, PI3K signaling rebound was efficiently blocked by co-targeting p110 α and p110 β ". However, the figure shows that in CWR22Pc-sgPTEN some rebound of pAKT (T308 as well as S473) is evident even with combination treatment with BYL719 plus AZD8186. Was this reproducibly observed? If so, this should be commented upon. This finding would be consistent with the overall hypothesis that direct AKT inhibition would be superior to p110 inhibition in the setting of PTEN loss.

Reviewer #2:

Remarks to the Author:

This is an interesting manuscript that addresses the important but complex issue of PI3K isoform dependency in PI3K pathway-altered prostate cancers. The authors use a broad collection of prostate cancer cell lines and patient-derived prostate cancer organoid models to show that PI3K isoform dependency is not governed by PTEN loss, as previously thought, but is impacted by feedback inhibition and concurrent p110 α and p110 β alterations. Their findings reveal a potential mechanism of resistance to PI3K inhibition and point to the need for different therapeutic strategies to target PI3K signaling in the context of different PI3K alterations.

The manuscript is well written and the data clearly presented. While it is perhaps not a definitive study, the work does provide an improved understanding of the mechanisms driving PI3K isoform dependency in prostate cancer and reveals important insights that will influence the design of future clinical trials.

Minor issues

Fig.1B. The key for "mRNA Upregulation" appears to be missing (presumably it is the pink?).

Fig.1H. PTEN was expressed using a Dox-inducible system (line 145). Does Fig.1H show LNCaP-PTEN cells with and without Dox induction, or LNCaP and LNCaP-PTEN cells (in which case, were both cell lines treated with DOX)? This should be clarified in the legend and image. The concentration of Dox used should also be provided.

Fig.2A-C,H. DDK? Presumably, the ectopically expressed p110 proteins are tagged with DDK? This

should be clarified in the legend. Also, the antibodies used for the p110 α and p110 β blots are not listed in the "Antibodies" section of the Methods (pages 23-24).

Fig.S2B. The legend says these are "CWR22Pc-sgPTEN-p110 β E1051K" cells. Is this correct? The heading on the image suggests they are "CWR22Pc-p110 β E1051K" cells (which would be more consistent with the text (Lines 178-180)).

Fig.4C. Should the blot labelled "IGFR1/INSR" be "pIGFR1/INSR"?

Line 494. "immunoblots were run in triplicate" – Does this mean the same sample blotted 3 times, 3 replicates from the same experiment, or 3 independent biological replicates (i.e. 3 different experiments)? Please clarify.

Reviewer #3:

Remarks to the Author:

This study by Mao et al examined the effects of PIK3CA/PIK3CB in AKT signaling and cell viability (in vitro) and tumor growth (in vivo) in the context of PTEN-negative or PTEN-positive condition. The initial cell models include 2 prostate cancer cell lines and 8 organoids. The authors attempted to address an interesting and clinically relevant question regarding the use of p110 isoform specific inhibitors in treating prostate cancer, which is a disease with frequent alterations in the PI3K pathway. In contrast to a previous study suggesting p110beta dependency in PTEN loss prostate tumor, by using more cell lines and organoids this study revealed that AKT phosphorylation in PTEN loss tumors may indeed depend on p110alpha or is independent of either one of them (Fig. 1F). The determining factor underlying this difference in dependency was not addressed. Some of the other data are less convincing or were not explained well. Below are the specific points:

(1) The authors have a rather wide panel of organoids but not all of them were used in the experiments. What are the rationales of picking which one to be used in each experiment? What are the mutation status of the other PI3K pathway members of these organoids except CA, CB, R1 and PTEN?

(2) The cell models do not carry intrinsic CA/CB mutations and therefore the mutants were overexpressed ectopically. AKT phosphorylation was increased by the overexpression and was reversed by the corresponding inhibitor, simply indicating that the increased AKT is mediated by the p110 isoform introduced. But the cell viability data did not align with the signaling (e.g. Fig. 2E BYL719 in p110 α E545K did not decrease cell viability), arguing a true therapeutic dependency. Also in Fig. 2G, either BYL719 or AZD8186 could decrease the cell viability, suggesting that both isoforms were involved instead of a simple shift in dependency?

(3) In Fig. 2E, why couldn't the expression of CA or CB oncogenic mutant enhance cell viability?

(4) In Fig. 2H, BYL719 or AZD8186 reduced AKT phosphorylation after WT p110 α expression. This observation is different from the expression of p110 α E545K as in Fig. 2D; this was not followed up.

(5) The cell models have endogenous PIK3R1 mutations which represent potential good models to study the effects of the mutations. The authors initially mentioned that R1 mutations are one of the common alterations in prostate cancer (Fig. 1A-D). Why was this gene dropped out in the subsequent experiments?

(6) In Fig. 3E, pAKT S473 could be suppressed by combined BYL719 and AZD8186 in vitro, but such combination failed to induce such suppression in vivo (Fig. 3H). Can the author explain this?

(7) What is the scientific ground of examining IGF1R at the first place out of the so many RTKs? Actually IGF1R has been shown to confer resistance to p110 specific inhibitors in multiple previous studies, e.g. "IGF1R upregulation confers resistance to isoform-specific inhibitors of PI3K in PIK3CA-driven ovarian cancer. Cell Death Dis. 2018".

"Activation of IGF1R/p110 β /AKT/mTOR confers resistance to α -specific PI3K inhibition. Breast Cancer Research. 2016"

Minor-

- In some experiments, phosphorylated S6 and/or PRAS40 was shown without the corresponding total proteins.

- Fig.1B, what do pink open rectangles indicate?

REVIEWER COMMENTS

Reviewer #1 (Remarks to the Author):

The manuscript by Mao et al. convincingly shows that PI3K p110 isoform dependency in prostate cancer is not simply driven by loss of PTEN, and gain-of-function mutations in PIK3CA or PIK3CB can confer or drive isoform dependency. Combined inhibition of both alpha and beta p110 isoforms is superior to selective inhibition, but resistance to p110 inhibitor combination treatment can be mediated by upregulation of IGF1R.

Taken together, the authors' findings show that direct inhibition of AKT may be superior to PI3K inhibition in PTEN-deficient tumors, whilst in PTEN wild-type tumors inhibition of p110 alpha is superior, due to simultaneous suppression of MAP kinase signaling.

These findings advance our understanding of the biology of PI3K-AKT signaling in distinct subsets of prostate cancers and provide a basis to explore differential therapeutic strategies based on biomarkers that are readily quantifiable.

The experimental approach is generally robust and the statistical methods used are appropriate. We appreciate the positive comments from reviewer #1.

Addressing the following comments will further strengthen the manuscript:

1. Figure 2H: ectopic overexpression of wild type p110alpha in LNCaP cells should be documented by Western blotting

We have labeled ectopic overexpression of wild type p110alpha in both figures through western blotting for DDK tag and total p110 \$\alpha\$ and reported this in the corresponding figure legends.

2. Figure 3F: The text states that "In both CWR22Pc-sgNT and -sgPTEN cells, PI3K signaling rebound was efficiently blocked by co-targeting p110 α and p110 β ". However, the figure shows that in CWR22Pc-sgPTEN some rebound of pAKT (T308 as well as S473) is evident even with combination treatment with BYL719 plus AZD8186. Was this reproducibly observed? If so, this should be commented upon. This finding would be consistent with the overall hypothesis that direct AKT inhibition would be superior to p110 inhibition in the setting of PTEN loss.

We thank reviewer #1 for highlighting this finding. A rebound of pAKT (T308 and S473) was indeed reproducibly seen in at least three independent repeats across several independent models, indicating that in a PTEN-deficient background, even the combination treatment of BYL719 plus AZD8186 was not sufficient to completely block PI3K signaling rebound. This is consistent with our overall hypothesis that direct AKT inhibition is superior as compared to PI3K inhibition in a PTEN deficient background. We have addressed this finding in the manuscript text.

Reviewer #2 (Remarks to the Author):

This is an interesting manuscript that addresses the important but complex issue of PI3K isoform dependency in PI3K pathway-altered prostate cancers. The authors use a broad collection of prostate cancer cell lines and patient-derived prostate cancer organoid models to show that PI3K isoform dependency is not governed by PTEN loss, as previously thought, but is impacted by feedback inhibition and concurrent p110 α and p110 β alterations. Their findings reveal a potential mechanism of resistance to PI3K inhibition and point to the need for different therapeutic strategies to target PI3K signaling in the context of different PI3K alterations.

The manuscript is well written and the data clearly presented. While it is perhaps not a definitive study, the work does provide an improved understanding of the mechanisms driving PI3K isoform dependency in prostate cancer and reveals important insights that will influence the design of future clinical trials.

We thank reviewer #2 for the encouraging comments. Indeed, we plan to further expand our preclinical findings with future clinical studies. This is currently being done through a collaborative effort with our Medical Oncology colleagues to develop clinical trials aimed at evaluating selective PI3K/AKT targeted therapeutic strategies in specific contexts of PI3K pathway altered prostate cancers based on our data.

Minor issues

Fig.1B. The key for “mRNA Upregulation” appears to be missing (presumably it is the pink?). Thank you for spotting this missing label, we have added the key for “mRNA Upregulation” to Fig. 1B.

Fig.1H. PTEN was expressed using a Dox-inducible system (line 145). Does Fig.1H show LNCaP-PTEN cells with and without Dox induction, or LNCaP and LNCaP-PTEN cells (in which case, were both cell lines treated with DOX)? This should be clarified in the legend and image. The concentration of Dox used should also be provided.

In Fig. 1H, LNCaP (left 3 lanes) and LNCaP-PTEN (right 3 lanes) cells were used. Doxycycline (500ng/mL) was added to all conditions for 8 hrs to induce PTEN expression or control vector, followed by AZD8186 or BYL719 treatment for 4hrs before cell lysates were collected. We added further descriptive details to Fig. 1H, corresponding figure legends and the method section.

Fig.2A-C,H. DDK? Presumably, the ectopically expressed p110 proteins are tagged with DDK? This should be clarified in the legend. Also, the antibodies used for the p110 α and p110 β blots are not listed in the “Antibodies” section of the Methods (pages 23-24).

We have added the details about the use of DDK-tagged p110 cDNAs in the figure legend. We also added antibody information in the method section for both p110 α and p110 β .

Fig.S2B. The legend says these are “CWR22Pc-sgPTEN-p110 β E1051K” cells. Is this correct? The heading on the image suggests they are “CWR22Pc-p110 β E1051K” cells (which would be more consistent with the text (Lines 178-180)).

We have fixed the figure legend to CWR22pc-p110 β ^{E1051K}.

Fig.4C. Should the blot labelled “IGFR1/INSR” be “pIGFR1/INSR”?

We have fixed the label to pIGFR1/INSR in Fig.4C. Total levels of IGF1R/INSR correlate well with its activity as indicated by corresponding pIGF1R/INSR level across multiple PDO lines (Fig. S4A).

Line 494. “immunoblots were run in triplicate” – Does this mean the same sample blotted 3 times, 3 replicates from the same experiment, or 3 independent biological replicates (i.e. 3 different experiments)? Please clarify.

We have provided clarifications to the corresponding method section.

Reviewer #3 (Remarks to the Author):

This study by Mao et al examined the effects of PIK3CA/PIK3CB in AKT signaling and cell viability (in vitro) and tumor growth (in vivo) in the context of PTEN-negative or PTEN-positive condition. The initial cell models include 2 prostate cancer cell lines and 8 organoids. The authors attempted to address an interesting and clinically relevant question regarding the use of p110 isoform specific

inhibitors in treating prostate cancer, which is a disease with frequent alterations in the PI3K pathway. In contrast to a previous study suggesting p110 β dependency in PTEN loss prostate tumor, by using more cell lines and organoids this study revealed that AKT phosphorylation in PTEN loss tumors may indeed depend on p110 α or is independent of either one of them (Fig. 1F). The determining factor underlying this difference in dependency was not addressed.

We appreciate that all the reviewers (including reviewer #3) acknowledged that PI3K signaling dependency is clinically relevant but biologically complex. There are clearly multi-layers of molecular mechanisms that could selectively drive PI3K dependency across different models/cancer types. For example, in a prior study that we have contributed to, prostate-specific membrane antigen (PMSA) or folate hydrolase 1 (FOLH1) could activate p110 β through G-protein coupled receptors independent on PTEN status (Kaittanis C, J Exp Med. 2017 PMID: 29141866). The goal of this manuscript is to focus on identifying clinically actionable mechanisms of p110 dependency and develop corresponding targeted therapeutic strategies to optimize response. We anticipate following studies from our group will further address context dependent mechanisms of p110 dependency from a more molecular mechanistic standpoint.

Some of the other data are less convincing or were not explained well. Below are the specific points: Please see following point-to-point response.

(1) The authors have a rather wide panel of organoids but not all of them were used in the experiments. What are the rationales of picking which one to be used in each experiment? What are the mutation status of the other PI3K pathway members of these organoids except CA, CB, R1 and PTEN?

In our study we have evaluated the impact of PI3K and AKT inhibition across our panel of prostate cancer organoids and cell lines (Fig 5F, G, and H) and have specifically devolved deeper into selective contexts throughout our experiments. Alterations in other members of PI3K pathway are also observed in CRPC patients (cbioportal.org) with lower frequency. We focused our study on the dominant drivers of PI3K signaling in prostate cancer (PTEN loss, PIK3CA/B mutation) where clinical grade inhibitors of the pathway are available. Among 4 members of gene encoding p110 proteins, PIK3CA and PIK3CB are highly expressed in epithelial cancer as compared to PIK3CD and PIK3CG (immune cells).

(2) The cell models do not carry intrinsic CA/CB mutations and therefore the mutants were overexpressed ectopically. AKT phosphorylation was increased by the overexpression and was reversed by the corresponding inhibitor, simply indicating that the increased AKT is mediated by the p110 isoform introduced. But the cell viability data did not align with the signaling (e.g. Fig. 2E BYL719 in p110 α E545K did not decrease cell viability), arguing a true therapeutic dependency. Also in Fig. 2G, either BYL719 or AZD8186 could decrease the cell viability, suggesting that both isoforms were involved instead of a simple shift in dependency?

We directly measure of PI3K activity, by blotting for pAKT (T308 and S473 sites) levels, which shows corresponding dependency to AZD8186 or BYL719 with either CA/CB ectopic expression in both LNCaP and CWR22Pc cells (Fig 2D, 2F). Therefore, introducing either CA or CB activating mutation by ectopic expression does promote p110 signaling dependency. Although our presented data showed the viability of LNCaP-p110 α ^{E545K} cells is slightly (but significantly) decreased by standard dose of BYL719 (1 μ M), these cells are sensitive to high dose of BYL719 (5 μ M) (Fig. S3D). Given that the p110 α ^{E545K} isoform is exogenously overexpressed, our data suggests that higher doses of BYL719 are necessary to achieve a dose to target ratio that is sufficient for inhibition in this context. Conversely, LNCaP-p110 α ^{E545K} cells remain completely resistant to higher dose of AZD8186 (1 μ M) (data not shown). The key point that our LNCaP data delivers is that introducing p110 α ^{E545K} abolishes its sensitivity to p110 β inhibition and thus switches the intrinsic p110 β -dependency towards a more dependency on p110 α selective inhibition, although signaling through both isoforms is relevant in the

setting of relief of feed-back rebound activation. Similarly, introducing p110 β ^{E1051K} into the p110 α dependent line CWR22Pc (CWR22Pc-p110 β ^{E1051K}) makes it equally sensitive to both AZD8186 and BYL719, indicating indeed there is a dependency shift from p110 α to p110 β . In addition, it is also known that p110 α activation is driven by growth factor-receptor tyrosine kinase mediated signaling cascade while p110 β activation is thought driven by G-protein coupled receptor, the latter being much weaker (Kaittani C, J Exp Med. 2017 PMID: 29141866). Therefore, oncogenic activity of p110 α is often much stronger than p110 β , and is likely to provide a more robust growth advantages. This is also consistent in our hands across different cell line and organoid models. The key conclusion from our data is that introduction of p110 α or p110 β activating mutants shift p110 signaling dependency by measuring its direct activity output (pAKT), while it also shifts sensitivity to either AZD8186 or BYL719 in cell proliferation assays in a context dependent manner.

We acknowledged some limitations of the model used with ectopic mutant PIK3CA/CB expression, which does not delineate the exact differences between PIK3CA/CB mutations versus amplification/up-regulation. Importantly, we have shown that overexpression of wild type p110 α does drive p110 isoform signaling dependency, although less potently compared to overexpression of an activating mutation. This is likely due to the overexpression of wild-type p110 α is still dependent on upstream RTK signaling input and thus downstream signaling is less robust compared to the activating mutations. In CRPC patients, some of the tumors harboring PIK3CA or PIK3CB mutations also simultaneously have amplification or mRNA up-regulation. Thus, our modeling approach will certainly reflect some if not all genetic makeup of CRPC patients (see below).

(3) In Fig. 2E, why couldn't the expression of CA or CB oncogenic mutant enhance cell viability? While expression of CA or CB does not have an obvious effect in enhancing cell viability in PTEN-deficient context (Fig. 2E), it does enhance cell viability in PTEN-wild type context (Fig. 2G). This is likely because the proliferative output from high PI3K signaling in PTEN-deficient cells is more saturated. While introduction of the PIK3CA or CB mutants does not have an obvious growth promoting effect to these already fast-proliferating PTEN-deficient cells, signaling dependency and therapeutic selectivity is impacted. Importantly, in PTEN-wild type cells, these mutants further enhanced CA or CB signaling can translate into enhanced proliferation.

(4) In Fig. 2H, BYL719 or AZD8186 reduced AKT phosphorylation after WT p110 α expression. This observation is different from the expression of p110 α E545K as in Fig. 2D; this was not followed up. Overexpression of WT p110 α was able to modify p110 isoform dependency, as shown in Fig. 2H. However, the effect is not as robust as when overexpressing p110 α E545K, indicating activating mutations of p110 α indeed play a dominant role in shifting p110 isoform dependency, which is not solely due to increase protein levels. Although this is not the focus of our current manuscript, the difference is intriguing and warrants future investigations.

(5) The cell models have endogenous PIK3R1 mutations which represent potential good models to study the effects of the mutations. The authors initially mentioned that R1 mutations are one of the common alterations in prostate cancer (Fig. 1A-D). Why was this gene dropped out in the subsequent experiments?

While the purpose of Figure 1A-D is to demonstrate the wide-spread alterations of key regulators of PI3K pathway in CRPC patients, the focus of our manuscript is to investigate the role of alterations of catalytic subunit of PI3K in signaling dependency and therapeutic response. This is of important clinical relevance as both p110 α and p110 β isoform-selective inhibitors are currently moving fast in clinical development. We have ongoing work with a collaborator focusing solely on functional dissection of alterations in the PI3K regulatory subunits (Xie et al., unpublished) which is molecularly complex and outside the bounds of our current study.

(6) In Fig. 3E, pAKT S473 could be suppressed by combined BYL719 and AZD8186 *in vitro*, but such combination failed to induce such suppression *in vivo* (Fig. 3H). Can the author explain this?

We appreciate reviewer #3's careful examination of our data presented in Fig. 3E and Fig. 3H and the observed difference. In the context of a PIK3CA mutant PTEN wild-type model rebound signaling through p110 β following p110 α is minimal, although it can be repressed. In this context p110 α is dominant as displayed in our *in vitro* and *in vivo* modeling, where there is no additive effect of AZD8186 to BYL719. Several presumptive factors may play into the observation made by the reviewer. First, the pharmacodynamics and pharmacokinetics are very different *in vitro* and *in vivo*, resulting in different dosage-dependent effects. Second, the analysis of *in vivo* tumors results in a pooled evaluation of tumor and microenvironment cells that may be differentially responsive to PI3K pathway inhibition. Third, while we have a defined media condition for cell culture, the tumor microenvironment is far more complex and different from the *in vitro* culture condition. It is also possible that different growth factors could activate the PI3K pathway despite the presence of inhibitors. Indeed, recent studies from the Cantley group have shown that serum insulin level could activate PI3K despite sustained inhibition of p110 α by BYL719 and thus limits its therapeutic efficacy (Hopkins et al., Nature 2018).

(7) What is the scientific ground of examining IGF1R at the first place out of the so many RTKs? Actually IGF1R has been shown to confer resistance to p110 specific inhibitors in multiple previous studies, e.g. "IGF1R upregulation confers resistance to isoform-specific inhibitors of PI3K in PIK3CA-driven ovarian cancer. Cell Death Dis. 2018".

"Activation of IGF1R/p110 β /AKT/mTOR confers resistance to α -specific PI3K inhibition. Breast Cancer Research. 2016"

Consistent with what reviewer #3 has mentioned, we are also aware of prior studies regarding activation of IGF1R could promote resistance to PI3K inhibitors in other type of cancers. We also have in house generated RNA sequencing data showing that among RTK members, IGF1R expression is selectively higher in PDOs that are resistant to BYL719 + AZD8186 combination. This makes IGF1R a natural candidate to test in our model systems.

Minor-

- In some experiments, phosphorylated S6 and/or PRAS40 was shown without the corresponding total proteins.

We have added total proteins level in Fig. 2H, 3A, 3C, 4C, 4G, 5E, S3A, S3C, S3E. These total protein levels remained unchanged across different conditions within our experiments.

- Fig.1B, what do pink open rectangles indicate?

Pink open rectangles indicate "mRNA Upregulation", we have fixed the Fig.1B with the missing key.

Reviewers' Comments:

Reviewer #1:

Remarks to the Author:

The revised manuscript completely addresses my concerns and would be of significant interest for the readers of Nature Communications due to its major mechanistic and translational implications.

Reviewer #2:

Remarks to the Author:

The authors have adequately addressed by comments.

Reviewer #3:

Remarks to the Author:

The replies of the authors are appreciated. Some questions have been clarified. However, some still remain. And many explanations in the rebuttal letter, which indeed provide useful information to the readers, are not incorporated in the revised manuscript.

The numbers below refer to my original comments in the first-round review.

(1) My original comment was asking about the mutation status of the PI3K pathway members of the organoids and cell lines used in this study. In Figure 1D, the authors show that of PIK3CA, PIK3R1 and PTEN. How about the other members? This question is highly relevant because this study involves PIK3CB and the AKT pathway.

(2) The difference in cell inhibition by the low and high BYL719 doses should be explained immediately when the Fig.2E data are mentioned in the text. The authors are suggested to supplement the text.

Also the authors replied that "Conversely, LNCaP-p110 α E545K cells remain completely resistant to higher dose of AZD8186 (1 μ M) (data not shown)." Can the data be included in the manuscript to better support the dependency?

(3) Again, this difference in cell numbers induced by expression of CA or CB in PTEN-deficient and PTEN-intact cells should be mentioned in the text. Also, statistically analysis should be done in Fig. 2G to see whether the enhanced cell number is significant.

(4) This observation should be mentioned in the text.

(7) The authors replied "We also have in house generated RNA sequencing data showing that among RTK members, IGF1R expression is selectively higher in PDOs that are resistant to BYL719 + AZD8186 combination." Can the authors show the expression data of some other RTKs that were not changed (along with that of IGF1R) to show the selectivity?

And it is fair to cite the two papers mentioned.

REVIEWER COMMENTS

Reviewer #1 (Remarks to the Author):

The revised manuscript completely addresses my concerns and would be of significant interest for the readers of Nature Communications due to its major mechanistic and translational implications.

Reviewer #2 (Remarks to the Author):

The authors have adequately addressed by comments.

We thank reviewers 1 and 2 for their positive comments.

Reviewer #3 (Remarks to the Author):

The replies of the authors are appreciated. Some questions have been clarified. However, some still remain. And many explanations in the rebuttal letter, which indeed provide useful information to the readers, are not incorporated in the revised manuscript.

The numbers below refer to my original comments in the first-round review.

(1) My original comment was asking about the mutation status of the PI3K pathway members of the organoids and cell lines used in this study. In Figure 1D, the authors show that of PIK3CA, PIK3R1 and PTEN. How about the other members? This question is highly relevant because this study involves PIK3CB and the AKT pathway.

We appreciate the reviewer's thoughtful insight and critiques to strengthen the concepts of our manuscript. We have included the data from our organoids regarding alterations involving PIK3CB and AKT. While there are low frequency mutations in PIK3CB and AKT in castrate-resistant metastatic prostate cancer, there were no alterations of PIK3CB or AKT in our patient derived prostate cancer organoids. We have clarified this in the text.

(2) The difference in cell inhibition by the low and high BYL719 doses should be explained immediately when the Fig.2E data are mentioned in the text. The authors are suggested to supplement the text. Also the authors replied that "Conversely, LNCaP-p110 α E545K cells remain completely resistant to higher dose of AZD8186 (1 μ M) (data not shown)." Can the data be included in the manuscript to better support the dependency?

We have made the changes as suggested by the reviewer by moving up a part of the discussion of high dose BYL719 in the LNCaP-p110 α E545K cells by referencing the data earlier in Fig 2E and moving the

figure to Supplementary Data 3A. We have also included the high dose AZD8186 data to demonstrate that the LNCaP-p110 α E545K cells are resistant to high dose p110b inhibition (Supplementary Fig 2A).

(3) Again, this difference in cell numbers induced by expression of CA or CB in PTEN-deficient and PTEN-intact cells should be mentioned in the text. Also, statistically analysis should be done in Fig. 2G to see whether the enhanced cell number is significant.

We have addressed this in the text and demonstrate that in the setting of wild-type PTEN, over-expression of the PIK3CA/PIK3CB mutants significantly promotes cell proliferation. Statistical analysis included.

(4) This observation should be mentioned in the text.

We have included a discussion in text regarding the signaling differences and signaling dependency between wild-type and PIK3CA mutant expression.

(7) The authors replied “We also have in house generated RNA sequencing data showing that among RTK members, IGF1R expression is selectively higher in PDOs that are resistant to BYL719 + AZD8186 combination.” Can the authors show the expression data of some other RTKs that were not changed (along with that of IGF1R) to show the selectivity?

We have included the RTK expression data form our prostate cancer organoids in supplemental data, supplementary Fig 4a, as well as in the text.

And it is fair to cite the two papers mentioned.

We have cited these two papers.

Reviewers' Comments:

Reviewer #3:

Remarks to the Author:

My concerns have been addressed.